# Induced endometrial trauma (endometrial scratch) in the mid-luteal menstrual cycle phase preceding first cycle IVF/ICSI versus usual IVF/ICSI therapy: study protocol for a randomised controlled trial

Clare Pye,[1] Robin Chatters,[2] Judith Cohen,[3] Kate Brian,[4] Ying C Cheong,[5] Susan Laird,[6] Lamiya Mohiyiddeen,[7] Jonathan Skull,[8] Stephen Walters,[9] Tracey Young,[9] Mostafa Metwally[8]

For numbered affiliations see end of article.

**Correspondence to**
Clare Pye; clare.pye@sth.nhs.uk

## ABSTRACT

**Introduction** Endometrial trauma commonly known as endometrial scratch (ES) has been shown to improve pregnancy rates in women with a history of repeated implantation failure undergoing in vitro fertilisation (IVF), with or without intracytoplasmic sperm injection (ICSI). However, the procedure has not yet been fully explored in women having IVF/ICSI for the first time. This study aims to examine the effect of performing an ES in the mid-luteal phase prior to a first-time IVF/ICSI cycle on the chances of achieving a clinical pregnancy and live birth. If ES can influence this success rate, there would be a significant cost saving to the National Health Service through decreasing the number of IVF/ICSI cycles necessary to achieve a pregnancy, increase the practice of single embryo transfer and consequently have a large impact on risks and costs associated with multiple pregnancies.

**Methods and analysis** This 30-month, UK, multicentre, parallel group, randomised controlled trial includes a 9-month internal pilot and health economic analysis recruiting 1044 women from 16 fertility units. It will follow up participants to identify if IVF/ICSI has been successful and live birth has occurred up to 6 weeks post partum. Primary analysis will be on an intention-to-treat basis. A substudy of endometrial samples obtained during the ES will assess the role of immune factors in embryo implantation. Main trial recruitment commenced on January 2017 and is ongoing. Participants randomised to the intervention group will receive the ES procedure in the mid-luteal phase of the preceding cycle prior to first-time IVF/ICSI treatment versus usual IVF/ICSI treatment in the control group, with 1:1 randomisation. The primary outcome is live birth rate after completed 24 weeks gestation.

**Ethics and dissemination** South Central—Berkshire Research Ethics Committee approved the protocol. Findings will be submitted to peer-reviewed journals and abstracts to relevant national and international conferences.

**Trial registration number** ISRCTN23800982; Pre-results.

### Strengths and limitations of this study

► This is the largest multicentre, pragmatic randomised controlled trial to date which aims to assess the effectiveness and cost-effectiveness of performing the endometrial scratch (ES) procedure in women having in vitro fertilisation (IVF)/intracytoplasmic sperm injection (ICSI) for the first time.
► It aims to determine whether performing an ES is an acceptable and well-tolerated procedure.
► Due to the nature of the intervention, it is not possible to blind study participants or clinicians.
► Potential difficulty with recruitment if patients are not in equipoise about effectiveness of the ES procedure in first-time IVF/ICSI cycles.

## BACKGROUND

The use of local endometrial trauma known as endometrial scratch (ES) to improve implantation rates in women undergoing assisted conception was first described in 2003.[1] The procedure has since been explored in several studies mainly focusing on women with recurrent implantation failure and has been shown to significantly increase pregnancy rates by almost double.[2–4] However, uncertainty remains as to the therapeutic effect of ES, due to heterogeneity of the populations included—and the timing and exact protocol of ES used—in previous evaluations.[5 6] Three systematic reviews have summarised the evidence, however, each included different studies.[2 7 8] A recent Cochrane review included 14 randomised studies: 7 in women with previous cycle failure, 5 in an unselected population and 1 in a first-time cycle.[8] The live birth rate (LBR) meta-analysis combined

trials regardless of the population (ie, number of previous in vitro fertilisation (IVF) cycles) and included five studies, reporting a risk ratio (RR) of 1.42 (95% CI 1.08 to 1.85), p=0.02. The odds of achieving a clinical pregnancy were also increased following ES with an RR of 1.34 (95% CI 1.11 to 1.62), p=0.002. The one trial conducted in women undergoing their first IVF cycle indicated the procedure was harmful with an OR of clinical pregnancy rate of 0.30 (95% CI 0.14 to 0.63), p=0.002.[9] Notably, this trial performed the ES procedure at the time of oocyte retrieval and not in the month prior to the IVF cycle. Despite the concerns around the quality of evidence in using ES and that many of the trials undertaken so far have been small (most <150 participants), ES has been widely adopted into routine clinical practice in women with recurrent unsuccessful implantation and is currently being provided in some fertility units where women are having IVF/intracytoplasmic sperm injection (ICSI) for the first time.[10 11] Two large trials are currently in progress to determine if ES is beneficial in women undergoing their second IVF cycle[12] and a sample of women undergoing any IVF cycle.[13] Therefore, given the lack of evidence for the effectiveness of ES in women undergoing their first cycle of IVF/ICSI, it is essential that a large well-controlled multicentre trial is conducted to fully investigate the effectiveness and safety of this technique.

The Human Fertilisation and Embryology Authority (HFEA) state in their statistical report into multiple births that the risks associated with multiple births is the single biggest health risk associated with fertility treatment.[14] Multiple births carry risks to the health of both the mother and the babies and that birth of a healthy singleton child, born at full term, is therefore the safest outcome of fertility treatment for both mother and child and is best achieved through promoting the practice of single embryo transfer (SET).

If ES can improve the implantation potential of the embryo and therefore improve success rates, ES may encourage an expansion of current SET policies. Inclusion of women with a lower chance of having cryopreserved embryos and a more general increase in the implementation of the practice of SET, could consequently have a large impact on the risks and costs associated with multiple pregnancies as a result of IVF.[15]

The exact mechanism by which ES may improve implantation is not yet known, however, it is known that implantation is a complex process involving a number of inflammatory mediators including uterine natural killer cells, leukaemia inhibitory factor and interleukin 15.[13] It is possible that ES may lead to the activation of inflammatory cells such as macrophages and dendritic cells, and release of inflammatory mediators such as tumour necrosis factor-α, interleukin-15, growth-regulated oncogene-α and macrophage inflammatory protein 1B.[14]

ES has also been shown to cause the modulation of several endometrial genes that may be involved in membrane stability during the process of implantation

such as bladder transmembranal protein (UPIb) and adipose differentiation-related protein and mucin 1.[16]

ES is routinely performed as an outpatient procedure. Risks have been identified in a previous study when the procedure was undertaken on the day of oocyte retrieval (reduced implantation and pregnancy rates)[9]; however, the procedure is not known to be associated with any particular risks when undertaken in the menstrual cycle preceding that of IVF therapy, apart from period like discomfort while performing the procedure. Taking simple analgesics prior to the procedure usually alleviates this. As with any intrauterine procedure, there is a potential for intrauterine infection. However, women attending for fertility treatment are usually screened for serious vaginal infections such as chlamydia to minimise the risk of any spread of infection when performing the embryo transfer procedure, a similar procedure to an ES as it involves the insertion of a catheter into the uterine cavity.

The main objectives of this trial are to assess the clinical and cost-effectiveness of the ES procedure in women aged between 18 and 37 years (inclusive) undergoing their first IVF/ICSI cycle using either antagonist or long protocols to see if it could potentially improve implantation rates and hence encourage the practice of single embryo replacement. A substudy will be undertaken in two of the fertility units where endometrial samples obtained from the ES procedure will be stored for later analysis to identify endometrial factors associated with successful pregnancy outcome.

## METHOD AND ANALYSIS

The ES trial is a multicentre, parallel group, randomised controlled trial to examine the clinical, cost-effectiveness and safety of an ES performed in the mid-luteal phase of the preceding cycle prior to a first-time IVF/ICSI cycle. Eligible participants will be randomised to either the treatment-as-usual (TAU) arm, consisting of usual IVF treatment, or the intervention arm where ES will be performed followed by usual IVF treatment. The overall study design is illustrated below in the study flow chart (figure 1).

### Patient and public involvement

The study was reviewed by couples waiting to commence IVF treatment and then by the members of The Jessop Wing Reproductive Health Public Advisory Panel, (patient and public involvement (PPI)) at the Jessop Wing, Sheffield. All were asked to provide input into the lay summary, recruitment strategy, visit schedule and benefits of the proposed study to the patient and the National Health Service (NHS). We asked about their experience of assisted conception, the things they liked and disliked, and the potential difficulties or barriers to attending for treatment, randomisation to the TAU arm and how this might affect recruitment but clarified that if the trial showed an increase in the scratch arm assisting embryo implantation, then it would form part

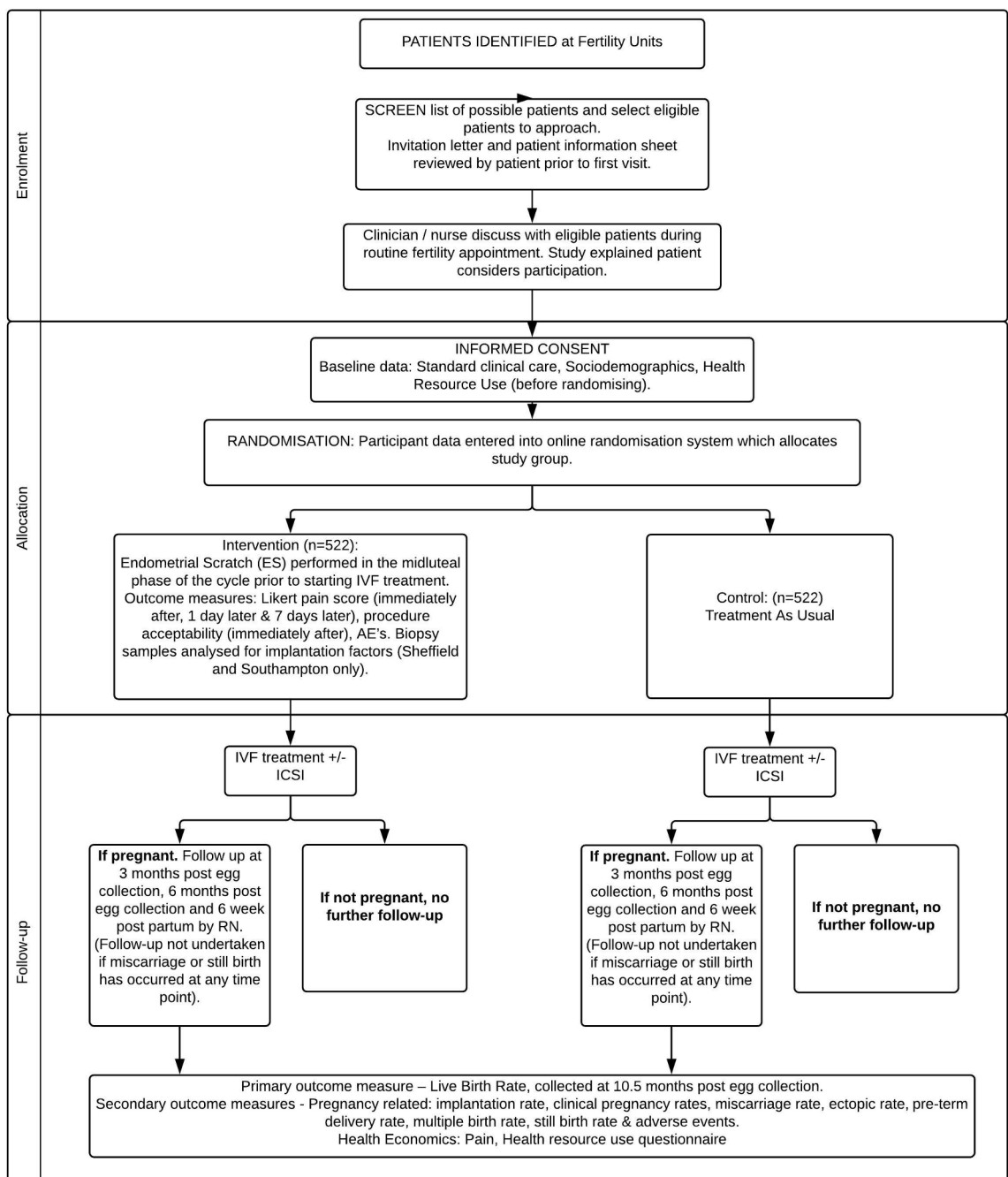

**Figure 1** Study flow chart. AEs, adverse events; ES, endometrial scratch; ICSI, intracytoplasmic sperm injection; IVF, in vitro fertilisation; RN, Research Nurse.

of the routine care pathway in the future. A member of the panel agreed to become the service user representative, is a member of the steering committee and attended the trial investigator/set-up meeting providing a patients view of all aspects associated with IVF. All PPI members have provided input into the patient facing documents on an ongoing basis and prior to submission to ethics and are aware of recruitment and the conduct of the study at ongoing PPI events held on a bimonthly basis within the directorate.

The most significant changes to the health technology assessment (HTA) grant influenced by the PPI members were in relation to trial follow-up procedures as they felt only women who achieved a pregnancy should be followed up. They also wanted to ensure continuity across the participating centres when performing the follow-up visits and requested a proforma be designed to ensure all research nurses/midwives capture the same information.

On completion of the trial, the results will be summarised in plain English and distributed to participants and patient support groups such as Infertility Network UK with the assistance of our service-user collaborators. We will promote the transfer of knowledge to wider audiences including the general public (eg, including short, user-friendly articles/briefings in relevant newsletters, magazines and periodicals, user groups/forums).

## Trial Design

The trial consists of two phases—an internal pilot to assess feasibility of recruitment and delivery of the intervention, and a 2-year main recruitment phase.

The trial will commence with a 9-month internal pilot recruitment phase across approximately six UK fertility units to justify whether or not the recruitment strategy and the scheduling of the ES procedure are feasible and will use the same trial procedures as described for the main trial.

The trial is collaboration between research staff at The Jessop Wing, Sheffield Teaching Hospital NHS Foundation Trust and The University of Sheffield—Clinical Trials Research Unit (CTRU) who is responsible for the conduct of the trial. Funding to run the trial has been awarded by the National Institute for Healthy Research HTA. At the end of the pilot phase, the trial steering committee (TSC) will report to the funder on whether the feasibility criteria have been met and whether the trial should continue.

The trial will be conducted in compliance with the approved protocol, Good Clinical Practice and regulatory requirements. Main trial recruitment commenced on January 2017 and is ongoing.

Sheffield CTRU will aggregate feasibility of the research and intervention protocols. The trial will be considered infeasible and will be stopped if either of the following conditions apply:

1. Feasibility of recruitment to the main trial: defined as recruitment of fewer than 108 participants (75% of the 144 target) during the internal pilot phase.
2. Scheduling of the ES procedure: defined as less than 75% of women scheduled to receive their ES procedure have received the ES at the correct time point.

## Recruitment

On successful completion of the pilot the main trial will aim to recruit women from 16 UK fertility units requiring first-time IVF treatment. Participation is entirely voluntary and choosing not to participate will not negatively influence the woman's treatment in any way. Furthermore consent can be withdrawn at any stage. Women who are about to undergo their first cycle of IVF/ICSI will be identified by screening patients referred for these treatments. Eligible women will be sent information regarding the trial in the post or via email or may be alerted to the trial via the trial website or posters displayed at the fertility unit. If they are interested in participating, they will be invited to discuss the trial with their fertility team at their next routine appointment.

Prior to randomisation full written informed consent will be obtained by a suitably trained doctor or research nurse/midwife at a clinic visit. The participant will complete a study-specific resource use questionnaire prior to randomisation to collect healthcare usage in the previous 3 months; baseline data will be collected at this visit and participants will be randomly allocated to either the intervention or usual care arm of the trial.

Detailed methods of the ES trial are described in the ES protocol available on the website (https://www.sheffield.ac.uk/scratchtrial).

Women will be included and considered suitable if they meet the following eligibility criteria:

### Inclusion criteria

1. Women expected to be aged between 18 and 37 years (inclusive) at time of egg collection.
2. First-time IVF with or without ICSI treatment using the antagonist or long protocol only.
3. Expected to receive treatment using fresh embryos.
4. Expected good responders to treatment, with:
   a. Ovulatory menstrual cycle (Regular menstrual cycles defined by clinical judgement or with ovulatory levels of mid-luteal serum progesterone as defined by local laboratory protocols).
   b. Normal uterine cavity (assessed by transvaginal sonography at screening and no endometrial abnormalities such as, suspected intrauterine adhesions, uterine septa, submucosal fibroids or intramural fibroids exceeding 4 cm in diameter as assessed by the investigator that would require treatment to facilitate pregnancy).
   c. Expected good ovarian reserve (assessed clinically, biochemically (follicle-stimulating hormone (FSH) <10 and normal follicular phase oestradiol levels and or normal anti-mullerian hormone (AMH)), and or sonographically (antral follicle counts) and no history of previous radiotherapy or chemotherapy). (All laboratory/ultrasound standards based on local normal reference ranges.)
   d. SET expected.
5. Local procedures have been/will be followed to exclude relevant vaginal/uterine infections prior to starting treatment.
6. Willing to use an appropriate method of barrier contraception if randomised to ES in the cycle where the ES procedure is performed.
7. Understands/willing to comply with the protocol.

### Exclusion criteria

1. Previous trauma/surgery to the endometrium (eg, resection of submucous fibroid, intrauterine adhesions).
2. Body mass index (BMI) of 35 kg/m$^2$ or greater.
3. Known grade 4 (severe) endometriosis.
4. Currently participating in any other fertility study involving medical/surgical intervention.
5. Expected to receive protocols other than antagonist or long (eg, ultralong protocol).
6. An ES (or similar procedure, eg, endometrial biopsy for the collection of natural killer cells) is planned.
7. Previously randomised into this trial.

### Sampling

The primary outcome is the LBR. This is defined as a live birth after completed 24 weeks gestation, within the 10.5-month post-egg collection follow-up period. This

time period will enable the collection of any neonatal deaths (up to 6 weeks post partum). The denominator for calculating the LBR will be the number of women randomised to each group. Data from the HFEA suggest an LBR of 32.8% in women under 35% and 27.3% in women aged 35–37. The sample size calculation assumes a 30% LBR in the control group and that an absolute increase of 10%, to a 40% LBR (a relative risk of 1.33) in the intervention groups is of clinical and practical importance. The effect size, a 10% absolute difference in LBR, we are proposing is large, but we believe an effect of such magnitude is needed to change clinical practice (there is a 5% absolute difference in LBR between women aged under 35 and 35–37) and is less than that observed in the systematic reviews described above (where, at the time of sample size calculation, the relative risk estimates for live birth ranged from 1.83 to 2.46).[2 17] To have a 90% power of detecting this difference or more, in LBR rates between the groups, as statistically significant at the 5% two-sided level, will require 496 women per group (992 in total). Adjusting for a predicted drop-out rate of 5% (due to anticipated difficulties of follow-up for patients who have been referred from NHS Trusts other than the participating fertility unit), we will require 1044 participants.

### Study procedures

Following randomisation women in the intervention arm will have the ES procedure performed in the mid-luteal phase of the preceding cycle prior to their planned IVF/ICSI cycle in the outpatient setting of the fertility unit. The choice of screening for infection prior to the procedure or the administration of antibiotics will be left to individual units according to their local established protocols and procedures. Women can be randomised any time up until they start their IVF cycle, although it may be necessary for the participant to delay her IVF if randomised to the intervention arm. This decision to delay should be made and agreed by both the patient and her fertility team before randomisation is undertaken. The contraceptive pill/oral contraceptives can be used following randomisation for the purposes of cycle programming, but women must be having ovulatory periods at the point of entry into the trial. Women randomised to TAU will continue with their IVF/ICSI as planned and will not receive the ES procedure.

Following delivery of the ES, participants will undergo IVF/ICSI in line with local procedures. Following successful embryo transfer (in both groups), a pregnancy test will be performed and adverse events (AEs) will be collected. In cases where women do not undergo embryo transfer, every effort will be made by the research team to collect any AE information from either the patient or the medical notes. If a pregnancy is confirmed, the woman is discharged to normal antenatal care as per standard practice. It is the intention to obtain the pregnancy status of all women once randomised including the outcome of all spontaneous pregnancies, the first frozen embryo transfer if no fresh transfer has been undertaken as well

as those that delay treatment following the ES. Women will not be followed up if they withdraw their consent from the trial. Data will also be collected regarding participants who have received an ES outside the trial.

### Randomisation

The randomisation schedule will be generated by Sheffield CTRU prior to the start of the trial; access to the schedule will be limited only to the trial statistician. The randomisation sequence will be computer generated and stratified by site and protocol (antagonist or long protocol). Random permuted blocks of variable size will be used to ensure enough participants are allocated evenly to each arm of the trial at each site. Research staff at recruiting centres will be unable to access the randomisation sequence and will use a web-based computer system with restricted access rights to enter participant details; randomisation outcome will then be revealed. Re-randomisation will not be permitted.

### Trial intervention

ES is a minor procedure of 10–20 min duration that will be performed in an outpatient setting at local IVF centres in line with local procedures and the trial standard operation procedure (SOP). The participant will be required to use a barrier method of contraception (if necessary) during the menstrual cycle in which the ES will be performed. During ES, a speculum is inserted into the vagina and the cervix exposed and cleaned. A pipelle or similar endometrial sampler is then inserted into the cavity of the uterus; negative pressure is applied by withdrawal of the plunger. The sampler is rotated and withdrawn several times so that tissue appears in the transparent tube. The sampler and speculum are then removed. If no tissue is seen in the transparent sampler, this is an indication that the sampler was not fully inside the uterine cavity and therefore the procedure is repeated. Following the procedure, women will complete a Visual Pain Scale (Likert) to assess their pain and tolerability assessment of the procedure within 30 min of the initial ES and then again at 24 hours and 7 days postprocedure via an automated text message.

Compliance to the intervention will be ascertained through the clinician or research nurse/midwife recording whether or not the patient has (1) attended the clinic for the ES procedure and (2) received the ES procedure as per protocol (PP). Any deviation from the protocol will be noted and reported as per the Sheffield CTRU SOP.

### Follow-up

Patient follow-up will continue until either the first cycle of IVF has been completed or the resulting pregnancy has concluded. If no pregnancy is confirmed, the study is complete (regardless of which group the woman is randomised to). Pregnant women will be followed up at 3 and 6 months post-egg collection and then 6 weeks post partum to collect pregnancy outcome and safety data. If the pregnancy is ongoing at 3 months and 6 weeks

post partum, a health resource use questionnaire will be sent to the patient for completion. If a spontaneous pregnancy is achieved between randomisation and IVF treatment, the pregnancy will be followed up as described above, instead of egg collection, the date of the last menstrual period will be used to schedule the 3/6 months and 6 weeks postpartum follow-ups.

### Safety considerations, safety monitoring and AE reporting

All AEs and serious AEs (SAEs) will be recorded by the local research team at each fertility unit. All AEs/SAEs will be followed up until satisfactory resolution or until the treating clinician and the principal investigator (PI) deems the event to be chronic or the participant to be stable. Research nurses/midwives will ask patients for any details of AEs at five time points: post procedure (if randomised to receive ES), at the participants' pregnancy test, and then, if pregnancy has been achieved, at 3 and 6 months post-egg collection and finally 6 weeks post partum.

AEs/SAEs will be collected up to the participants' final study-related follow-up event. If embryo transfer does not occur, the research nurse/midwife will contact the participant approximately 2 weeks after egg collection to identify if any AEs have occurred. In the case of a negative pregnancy test, the site research team should make every effort to obtain AE data from the patient or the medical notes at routine clinical care contacts; no further contact will be made outside of routine clinical care.

Expected AEs will be those which occur regularly due to pregnancy, and expected SAEs are those events which are expected in the patient population as a result of the routine care/treatment of a patient. Expected SAEs and all AEs will be collected as part of the trial and entered into the electronic case report form, but will not be reported to regulatory bodies (NHS REC/sponsor).

Unexpected SAEs will be reported to the Sheffield CTRU as soon as staff at the fertility units become aware of the event.

All SAEs will be reviewed by the data monitoring and ethics committee (DMEC) and trial management group (TMG) at regular intervals. The chief investigator will inform all PIs concerned of relevant information that would adversely affect the safety of the participants.

### Outcomes

#### Primary clinical outcome

► Live birth rate based on the number of live births after 24 weeks gestation within the 10.5-month post-egg collection follow-up period.

#### Secondary outcomes

► Acceptability and pain rating of the ES procedure, a Visual Pain Scale (Likert) to assess their pain and tolerability assessment of the procedure within 30 min of the initial ES procedure, 1 day later and then again 7 days after the ES.

► Implantation rate based on a positive serum beta human chorionic gonadotropin (hCG) on approximately day 14 following the egg collection or by a positive urine pregnancy test.

► Clinical pregnancy rate: an observation of viable intra-uterine pregnancy with a positive heart pulsation seen on ultrasound at/after 8 weeks gestation.

► Miscarriage rate as measured by spontaneous pregnancy loss (including pregnancy of unknown location prior to 24 weeks gestation within the 10.5-month post-egg collection follow-up period.

► Ectopic pregnancy as measured by the rate of pregnancy outside the normal uterine cavity.

► Multiple birth rate defined as the birth of more than one living fetus after completed 24 weeks gestation.

► Preterm delivery rate as measured by live birth after 24 weeks but before 37 weeks gestation within the 10.5-month postegg collection follow-up period.

► Stillbirth rate based on the delivery of a stillborn fetus showing no signs of life after 24 weeks gestation within the 10.5-month post-egg collection follow-up period.

► Details of participant's IVF cycles including number of eggs retrieved, number of embryos generated 1 day after egg collection, quality of the embryos transferred (using National External Quality Assessment Service (NEQAS) grading) and the number of embryos replaced and day of embryo replacement.

► AEs.

► Health resource use of the participant and patient costs.

The trial includes a health economic component to assess the cost of the intervention per extra live birth from an NHS and social care perspective. Resource use will include the intervention costs for ES, the cost of IVF treatment, visits to the Assisted Conception Unit and for those who conceive antenatal and postnatal visits, delivery costs and any hospital stays not related to birth for both mother and baby. The resource use questionnaire will collect information on contacts with midwife and general practitioner visits. A Patient Cost questionnaire will collect time taken to travel to appointments and loss of productivity. Unit costs will be derived from appropriate national sources and will include; NHS reference costs, Personal Social Service Research Unit costs and the Office of National Statistics.[18–20] The resource use questionnaire will be designed for this study and will draw on data collection tools developed in The School of Health and Related Research and those collated by the Database of Instruments for Resource Use Measurement.

### Blinding

Since this trial evaluates objectively measured outcomes (pregnancy rates) that are unlikely to be affected by a placebo effect, participants will not be blinded to treatment allocation; it is therefore not necessary to perform a sham procedure for the control group. The study statistician, TSC and health economist will be blinded to the allocation.

## Trial monitoring and oversight committees

The trial will be overseen by the TSC and the DMEC, membership of both will consist of independent experts in the field. The TSC will include a patient representative. Both committees will review recruitment, study progress and AEs. The DMEC will receive monthly reports of recruitment and AEs and, at their meetings, will also consider emerging evidence from other trials or research on ES. They may advise the chair of the TSC at any time if, in their view, the trial should be stopped for ethical reasons, including concerns about patient safety.

Day-to-day running of the trial will be coordinated by the TMG, consisting of the grant coapplicants, plus members of the Jessop Wing Fertility Unit, Sheffield CTRU and Patient representatives.

## Statistical analysis

Primary analysis will be performed on the intention-to-treat population (all participants randomised into the trial). All statistical exploratory tests will be two tailed at 5% nominal level. Baseline demographic (eg, age), physical measurements (eg, BMI) and health-related data will be described and summarised overall and for both treatment groups. The women, not the IVF cycle, will be the unit of analysis. If the woman fails to get pregnant or does not have IVF treatment, they will be included in the analysis of the primary outcome as a negative outcome (ie, non-live birth). For sensitivity analyses, PP analyses will also be undertaken which will be defined as for ES participants in the intervention group, receiving the ES procedure as documented in the study protocol and undergoing IVF/ICSI in the subsequent menstrual cycle, including embryo transfer. For the control group, the PP population will receive IVF/ICSI including embryo transfer. Subgroup analyses will be undertaken to explore the effect of important variables related to the participant and their treatment on the primary and secondary outcomes. These subgroups are:

► Day of embryo transfer (day 2, 3, 4, 5 or 6).
► Fertilisation method (IVF, IVF or ICSI, ICSI (spilt)).
► Type of protocol (long or antagonistic).
► Embryo transfer (single or double) and whether the embryo was fresh or frozen.
► Previous history of consecutive miscarriages (0–2 vs ≥3).

AEs will be reported as a proportion of all women randomised. AEs including SAEs will be compared between the two groups using a Fisher's exact test, $\chi^2$ test or negative binomial regression model in case of repeated events per woman (as appropriate). A 95% CI for the difference in AE rate between the groups will also be calculated with associated point estimate depending on the method used.

Health economic results will be presented in the net-benefit framework and will allow for uncertainty using bootstrapping and probabilistic sensitivity analysis.

## Ethics and dissemination

The findings of this trial will be submitted to peer-reviewed journals and abstracts to national and international conferences. Other stakeholder-specific outputs in relevant formats will also be produced for commissioners, IVF practitioners, third sector and user advocacy organisations. A website will be established to promote the work of the trial. All knowledge transfer activity including translation will be informed by input from trial collaborators, the TSC and TMG to ensure the study is meeting the needs of the commissioners and audience.

## DISCUSSION

This trial will determine whether performing an ES procedure prior to first-time IVF/ICSI treatment is an inexpensive, safe and well-tolerated procedure that increases the LBR in women having SET. If shown to be the case, this will have a significant improvement in first cycle IVF success rates and potentially lead to significant cost savings to the NHS as fewer women would need to have repeat treatment cycles. This is particularly important in the current economic climate and with restrictions on funding and service provision. This will also have a significant impact for women, for whom the burden of repeated cycles is large.

**Author affiliations**
[1]Sheffield Teaching Hospitals NHS Foundation Trust, Sheffield, UK
[2]Clinical Trials Research Unit, School of Health and Related Research (ScHARR), University of Sheffield, Sheffield, UK
[3]Hull Health Trials Unit, University of Hull, Hull
[4]Patient and Public Involvement (PPI), Fertility Network UK, London, UK
[5]University Hospital Southampton NHS Foundation Trust, Southampton, UK
[6]Biomolecular Sciences Research Centre, Sheffield Hallam University, Sheffield, UK
[7]Saint Mary's Hospital - Central Manchester University Hospitals, Manchester, UK
[8]Fertility Unit - Jessop Wing, Sheffield Teaching Hospitals NHS Foundation Trust, Sheffield, UK
[9]School of Health and Related Research (ScHARR), University of Sheffield, Sheffield, UK

**Acknowledgements** We thank Dr Munya Dimairo—research fellow in medical statistics at the University of Sheffield for his contribution to the design/content of the protocol. The Jessop Wing Reproductive Health Public Advisory Panel as well as each local PI, research nurse/midwife and research delivery team in the recruitment and management of patients.

**Contributors** MM, CP, JC, KB, YCC, SL, LM, JS, SW and TY conceived the study, and contributed to study design, sample size calculations and analytical plans. MM, CP, RC, JC, KB, YCC, SL, LM, JS, SW and TY initiated the project, assisted in developing the protocol and helped with implementation. CP, RC, JC and MM drafted the manuscript. All authors read and approved the final manuscript.

**Funding** This trial was funded by the NIHR Health Technology Assessment programme grant number (14/08/45). This article presents independent research supported by the National Institute for Health Research (NIHR).

**Disclaimer** The views expressed are those of the author(s) and not necessarily those of the NHS, the NIHR or the Department of Health.

**Competing interests** None declared.

**Patient consent** Not required.

**Ethics approval** The study is registered on the ISRCTN database (reference 23800982) and has been approved by the South Berkshire Research Ethics Committee (reference 16/SC/0151).

**Provenance and peer review** Not commissioned; externally peer reviewed.

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
