## [Reviewer comments · BMJ Open]

ARTICLE DETAILS

TITLE (PROVISIONAL)	Induced Endometrial Trauma (endometrial scratch) in the mid-luteal menstrual cycle phase preceding first cycle IVF/ICSI versus usual IVF/ICSI therapy: Study protocol for a randomised controlled trial.
AUTHORS	Pye, Clare; Chatters, Robin; Cohen, Judith; Brian, Kate; Cheong, Ying; Laird, Susan; Mohiyiddeen, Lamiya; Skull, Jonathan; Walters, Stephen; Young, T; Metwally, Mostafa

VERSION 1 – REVIEW

REVIEWER	Sarah Lensen University of Auckland, New Zealand
REVIEW RETURNED	30-Nov-2017

GENERAL COMMENTS	Thank you for the opportunity to review this manuscript for a randomised trial investigating the potential effect of endometrial scratching on women undergoing their first IVF cycle. The protocol explains well the rationale for the trial and the methodology; I have the following requests for minor corrections/clarification - There is too much emphasis on the impact of the trial results on the rate of SET vs DET in the future and possible impacts on multiple pregnancy rates. This is not a direct outcome of your study, perhaps it is best referenced more briefly and in the discussion.- In the background description of the evidence base, the Cochrane review is referenced but it does not mention the trial quality and issues of risk of bias – only trial size. The fact that the result of the meta-analysis is significant goes against the argument of trial size being an issue; trial quality is a big issue.- “Risks have been identified in a previous study when the procedure was undertaken on the day of oocyte retrieval” – what are the risks? Do you mean here a possible decrease in the chance of pregnancy? Other risks include pain (we might not have much of evidence about pain from trials of ES per se but it’s the same as an endometrial biopsy procedure) and inconvenience to the patient (attending for another appointment – will this be conducted with concurrent scans/appointments or not). Theoretical risks could be related to abnormal placentation.- Randomisation 4 months prior to consent introduces an unnecessary period for potential attrition and spontaneous pregnancies. Why not randomise closer to the time of the intervention?- How long will you wait for participants to have their IVF treatment following randomisation? If the participant decides to delay their cycle for any reason, and then they have their cycle 3 months later, 6 months later, a year later – will you count that? What if they have their scratch and then a 6 month break before the IVF? There should be a system which applies uniformly to all participants,
---

	regardless of their trial allocation. I see there is a 10.5 month rule for counting births post egg collection, but there is no description of how delays between randomisation and commencing IVF will be treated  - Following the above, it is not clear whether only the fresh embryo transfer will be counted or whether a frozen transfer would be counted in women having a freeze-all? I see there is subgroup analysis planned for whether the transfer is fresh or frozen so there may be some expectation that a women may manage to have a freeze-all and then frozen transfer, conceive and then have a live birth within the 10.5 months. Further, women may even manage to conceive spontaneously immediately following the IVF cycle (freeze-all or even after a fresh transfer) – would you count this spontaneous pregnancy if the birth occurred within 10.5 months? - The protocol is written in the language of “this will be done” when in reality the study has already been running for some time, not sure if this matters or not? - Inclusion criteria are not very specific, for example they state participants need to have “expected good ovarian reserve” – are any limits for bFSH, AFC or AMH provided around this? And that they should have a “normal uterine cavity” – would a small polyp still be normal? - Women must be using an antagonist or long agonist protocol – can either have pill start? - The exclusion criteria states “previous trauma/surgery to the endometrium and have a BMI of 35 kg/m2 or greater with known grade 4 (severe) endometriosis” this implies if they had surgery but not a BMI over 35 they would be eligible? These should all be separate bullet points I think for clarity - Women are excluded if they have previously had an endometrial biopsy for measure of NK cells – what if this was 5 years ago and not part of an IVF cycle? - Women may go and pay for the ES privately if in the control arm, how will you capture this and will you exclude these women from the PP analysis? - The discussion states that “This trial will determine whether performing an ES procedure prior to 1st time IVF/ICSI treatment is an inexpensive, safe and well tolerated procedure that increases the live birth rate in women having SET” – why mention SET? Some of your patients will have DET as you list this as a subgroup analysis. - I think a figure describing when the ES will happen relative to both the long agonist and antagonist protocols would be helpful (including the use of any pill-start that might be permitted).
--	--

REVIEWER	Wellington Martins SEMEAR fertilidade, Reproductive Medicine. Ribeirao Preto - SP, Brazil
REVIEW RETURNED	03-Dec-2017

GENERAL COMMENTS	Well written protocol assessing an important question. The only limitation is that the study will be powered only to detect differences > 10% in live birth rate, while a 5% difference (NNT = 20) would already be somewhat relevant, considering the cost/risks of the procedure and the total costs of IVF. However, I do understand that the required sample size to identify a 5% increase would be very large (2900 women should be randomized).
--

REVIEWER	Amerigo Vitagliano Department of Women' and Children's Health, Padua University, Italy.
-----------------	--

GENERAL COMMENTS

General comment:

Authors aim to evaluate the effectiveness of mid-luteal endometrial scratching in women undergoing their first IVF-ICSI cycle through a multi-center randomized controlled trial. The topic is of major interest and the study protocol is well designed. As Author correctly state, their study will be the largest randomized controlled trial on patients undergoing their first IVF attempt. Thus, it is expected to provide a powerful contribute for both scientists and fertility care providers about the practice of endometrial scratching.

Given the huge expectations linked to the trial in object, some minor changes to the protocol might be helpful to avoid methodological bias and to implement the reliability of Authors' results.

Specific comments:

1)Background, page 4, lines 13-34: Authors properly described the results of recent meta-analysis on endometrial injury before IVF, focusing on the findings provided by Nastri et al. In addition, they correctly mention the study by Karimzade et al, emphasizing the negative impact of endometrial scratching at the day of oocyte retrieval on clinical pregnancy rate. For completeness, Authors should specify that current evidence on the effects of endometrial scratching is considerably limited by heterogeneity among studies in terms of techniques (i.e. curette, pipelle, hysteroscope) and timing (i.e. Nastri et al study includes patients receiving endometrial scratching from the 7th day of the cycle before IVF to the day of oocyte retrieval). Moreover, they should mention that the practice of endometrial scratching is currently being evaluated also in other ARTs (i.e. intrauterine insemination and ovulation induction with sexual intercourse).

2)Inclusion criteria, page 8, lines 11-14: "women expected to receive treatment using fresh embryos and considered to be good responders to treatment (Regular ovulatory menstrual cycle, Normal uterine cavity, expected good ovarian reserve)". Authors should state how "normal uterine cavity" was assessed (i.e. with hysteroscopy or sonohysterography) and should provide more details about how ovulation was confirmed (otherwise they may simply state "history of regular menstrual cycle"), as well as criteria for defining "expected good ovarian reserve" (i.e. absence of Bologna criteria, precise cut-off of FSH and/or AMH). Moreover, Authors should specify if autoimmune disease, previous chemotherapy and endocrine disorders will represent additional exclusion criteria.

3)Randomisation, page 9, lines 32-36: "computer is certainly acceptable for generating a random sequence. Nevertheless, a strategy for allocation concealment should be specified in order to avoid bias due to inappropriate allocation (i.e. sealed envelopes, central allocation).

4)Trial intervention, page 9, lines 45-46: "A pipelle or similar endometrial sampler is then inserted into the cavity of the uterus". Authors should specify which kind of "endometrial sampler similar to pipelle" will be considered as appropriate in their study, in order to avoid performance bias between centers. Moreover, it would be important to ascertain that "mid-luteal scratching" is performed with comparable timing in all patients. Thus, Authors may clearly define when the procedure will be performed (i.e. from day LH + 6 to LH + 8 if ovulation is assessed by LH measurement, menstrual cycle length minus 5-9 days if ovulation will not be assessed etc)

5)Outcomes, page 11: Authors should provide a clear definition of secondary outcomes (i.e. clinical pregnancy rate will be defined as

	the visualization of fetal heart activity at ultrasound scan) and the denominator of the outcomes miscarriage rate, ectopic pregnancy rate, multiple pregnancy rate, preterm delivery rate, still birth rate (per patient/per pregnancy). 6)IVF cycles: Authors might describe more appropriately the criteria for ovulation induction (i.e. two or more follicles with mean diameter ≥ 18 mm) and specific drugs employed (i.e. 10,000 IU urinary hcg). In addition, as the primary outcome is related to pregnancy, a clear description of luteal phase support, if applicable, (i.e. vaginal progesterone, with dose) should be supplied.
--	--

REVIEWER	Ahmed Gibreel Mansoura University Egypt
REVIEW RETURNED	08-Dec-2017

GENERAL COMMENTS	This is a very well written protocol to elucidate the genuine effect of endometrial trauma on the outcome of IVF for women undergoing IVF/ICSI for the first time. My only concern about this study related to the population included. Women undergoing IVF for the first time have been consistently shown not to get benefit from the procedure in most of similar trials. A subgroup analysis of trials, where the population was women who underwent ≤ 1 previous attempt of embryo transfer, in the last update of the Cochrane review (Nastri et al, 2015). In that analysis, 4 trials were identified and included. Upon aggregating the data, the conclusion was "As the estimate was imprecise, it was not possible to ascertain whether this intervention was related to no effect or benefit: RR 1.1, 95% CI 0.91 to 1.33; P value 0.32; four RCTs; 650 women; $I^2 = 0\%$; low-quality evidence. This means no single randomised trial had shown the very high absolute difference expected in the protocol (increase live birth from 30% to 40% or an increase in the RR to 1.33). I am a bit concerned that authors may opt to choose this assumption to decrease the sample size needed. Justifications that only procedures that increase success rate by 33% would be worth changing the practice may be too difficult to be accepted. I would recommend recalculating sample size based on the assumption that the procedure would increase Live birth rate from 30% up to 36%. There are also some points in the protocol that need attention:  1) In the background section in page 5, line 19: the two references inserted [6,7], I could not figure out why putting one trial and one reference for a protocol published in clinical trials as reference for a meta-analysis . 2) The authors may want to mention the two large ongoing trials to elucidate the effect of scratch (van Hoogenhuijze et al, 2017) and Lansen et al, 2016).
---

REVIEWER	Carlos Simon Valencia University, Valencia, Spain
REVIEW RETURNED	11-Dec-2017

GENERAL COMMENTS	The methodological design of the paper is OK, however the nature of the complexity of this topic is not due to the lack of clinical data, since more than 1,000 papers, 15 RCTs and 5 meta-analyses have been already published. The main issue to be addressed is the fact that intentional damage to the endometrial lining known as endometrial scratching (ES) has been elevated to the category of therapeutic intervention to improve endometrial receptivity, but ES is not an intervention that has been
--

	or can be standardized. The present study, as all previous, lacks a reproducible definition and methodology of this “intervention” such as the type of catheter to be used, where in the endometrial cavity will be performed, how deep and/or extent the “curettage like” should be, (endometrial tissue obtained from the curettage should be analyzed to determine how deep the ‘intervention’ reached), US guided or not, at what exact day of the luteal phase will be done and why, etc. Even more worrisome is that the biological responses induced by ES that should be the scientific basis of this practice, remain uncertain and unproven. There is no precedent in any medical field that scratching a tissue or organ without knowing its effects has been elevated to the category of therapeutic intervention and is offering as a “therapy”. Minor issues: The background should reflect that no consensus exists about what scratching is, how to perform it and its "therapeutic" effect. Critical literature with the validity of this procedure should also be incorporated for a proper balance. -Simon C, Bellver J. Scratching beneath ‘The Scratching Case’: systematic reviews and meta-analyses, the back door for evidence-based medicine. Human Reprod 2014; 29:1618–1621. - Santamaria X, Katzorke N, Simon C. Endometrial ‘scratching’: what the data show Current Opinion Obgyn 4: 242-249, 2016
--	---

VERSION 1 – AUTHOR RESPONSE

Reviewer 1 - Sarah Lensen.

We would like to thank Dr Lensen for her comments; we also appreciate her declaring a conflict of interest. We were indeed approached by Dr Lensen’s group from New Zealand some time ago to contribute to their study but given that this would have a conflict of interest with this study we were unable to accommodate their requests. There is of course a potential continuing conflict of interest that we would like to highlight.

Q1. There is too much emphasis on the impact of the trial results on the rate of SET vs DET in the future and possible impacts on multiple pregnancy rates. This is not a direct outcome of your study, perhaps it is best referenced more briefly and in the discussion.

Response: The case for potential benefit of this study and promoting single embryo transfer verses double embryo transfer was a key part of our protocol including the economic analysis of the potential benefits. These were carefully considered by the HTA when peer reviewing our study and making the case for funding.

Q2. In the background description of the evidence base, the Cochrane review is referenced but it does not mention the trial quality and issues of risk of bias – only trial size. The fact that the result of the meta-analysis is significant goes against the argument of trial size being an issue; trial quality is a big issue.

Response: Regarding the Cochrane review we agree that current evidence shows a significant problem with quality of trials, however the Cochrane review focuses on women with recurrent implantation failure which is a different topic to our current study which focuses on women having

their first time IVF. We have reflected on the poor quality of current trials within the background section of the manuscript.

Q3. "Risks have been identified in a previous study when the procedure was undertaken on the day of oocyte retrieval" – what are the risks? Do you mean here a possible decrease in the chance of pregnancy? Other risks include pain (we might not have much of evidence about pain from trials of ES per se but it's the same as an endometrial biopsy procedure) and inconvenience to the patient (attending for another appointment – will this be conducted with concurrent scans/appointments or not). Theoretical risks could be related to abnormal placentation.

Response: Regarding the mention of studies which have a detrimental effect on the outcomes when the endometrial scratch is performed on the day of the egg collection, this was clear to the HTA that it referred to the one study that had been referenced which showed a decrease in pregnancy rates when the scratch was performed on the day of egg collection. We have clarified the risks within the manuscript. This was part of our balanced argument for the potential risks and benefits of endometrial scratch which were taken into careful consideration in the design of this study and the decision on funding.

Q4. Randomisation 4 months prior to consent introduces an unnecessary period for potential attrition and spontaneous pregnancies. Why not randomise closer to the time of the intervention?

Response: Our study procedure requires randomisation to occur as close to consent as possible and is generally performed at the same time. The 4month time period was included in the trial protocol for project management reasons. This currently isn't stated in the manuscript but only in the protocol and in the event of an amendment will make the following change to the protocol... ' Randomisation should be undertaken up to four months before the participant is due to commence her IVF therapy...'

Q5. How long will you wait for participants to have their IVF treatment following randomisation? If the participant decides to delay their cycle for any reason, and then they have their cycle 3 months later, 6 months later, a year later – will you count that? What if they have their scratch and then a 6 month break before the IVF? There should be a system which applies uniformly to all participants, regardless of their trial allocation. I see there is a 10.5 month rule for counting births post egg collection, but there is no description of how delays between randomisation and commencing IVF will be treated.

Response: The participant is able to delay their cycle for as long as they wish as long as the delay is within the project timelines. Women are expected to receive ES in the mid-luteal phase of the preceding menstrual cycle and if for some reason the scratch is performed and the IVF is delayed we will still continue to follow-up the outcome of the cycle once it is performed. No women are excluded from the study after randomisation unless they withdraw consent or the investigator withdraws them for a safety reason. Clarified in manuscript.

Q6. Following the above, it is not clear whether only the fresh embryo transfer will be counted or whether a frozen transfer would be counted in women having a freeze-all? I see there is subgroup analysis planned for whether the transfer is fresh or frozen so there may be some expectation that a women may manage to have a freeze-all and then frozen transfer, conceive and then have a live birth within the 10.5 months. Further, women may even manage to conceive spontaneously immediately following the IVF cycle (freeze-all or even after a fresh transfer) – would you count this spontaneous pregnancy if the birth occurred within 10.5 months?

Response: Our protocol states that our study is about women who expect to have single fresh embryo transfer. Of course having embryos frozen is a consequence of any IVF cycle – in this trial, the first FET will be followed up, if no fresh transfer has been undertaken already. The possible use of FET is

also accounted for in our analysis, which has been extensively discussed along the course of the study and our statisticians, who plan to account for this as part of the subgroup analysis and related exclusion criteria have been set for the per protocol analysis. This has been clarified in the manuscript.

Q7. The protocol is written in the language of “this will be done” when in reality the study has already been running for some time, not sure if this matters or not?

Response: Regarding the mention of the future tense when publishing the protocol – our aim here is to publish the protocol as it was decided at the start of the study so that our results can be validated against our protocol for the purpose of transparency. The protocol would therefore always be referred to in the future tense, however we have now included a sentence at the beginning to clarify that this study has in fact now started recruitment and has already been peer reviewed through the NHIR and that the aim of the publication is to ensure that this protocol is transparent and in the public domain for the future publication of our results.

Q8. Inclusion criteria are not very specific, for example they state participants need to have “expected good ovarian reserve” – are any limits for bFSH, AFC or AMH provided around this? And that they should have a “normal uterine cavity” – would a small polyp still be normal?

Response: For the purposes of the manuscript the word count was condensed but appreciate the reviewers’ comments and the manuscript has now been amended to show the full inclusion/exclusion criteria.

Q9. Women must be using an antagonist or long agonist protocol – can either have pill start?

Response: Some sites use OCP for the purposes of cycle programming so IVF can be planned which is fine for this study but women must be having ovulatory periods at the point of entry. This has been clarified in the manuscript.

Q10. Women are excluded if they have previously had an endometrial biopsy for measure of NK cells – what if this was 5 years ago and not part of an IVF cycle?

Response: Regarding Endometrial biopsy for natural killer cells – our study is about women having first time IVF, therefore there is no indication why these women would have had a natural killer cell biopsy in the past, since this would be done only if they had had several unsuccessful treatment cycles and therefore this question is not relevant to our current study.

Q11. Women may go and pay for the ES privately if in the control arm, how will you capture this and will you exclude these women from the PP analysis?

Response: We thank the reviewer for this comment which has been discussed within the trial management group and of course is an inevitable possibility which is captured in our data collection and is treated as a protocol violation and will be accounted for in the intention to treat analysis. This has now been added to the manuscript

Q12. The discussion states that “This trial will determine whether performing an ES procedure prior to 1st time IVF/ICSI treatment is an inexpensive, safe and well tolerated procedure that increases the live birth rate in women having SET” – why mention SET? Some of your patients will have DET as you list this as a subgroup analysis.

Response: The primary intention of the study is to examine women with single embryo transfer. This is however a pragmatic study and it is known that some patients will have a double embryo transfer and these will be analysed separately therefore, a subgroup analysis will be done to still answer the question whether this is a safe effective procedure in women with single embryo transfer.

Q13. I think a figure describing when the ES will happen relative to both the long agonist and antagonist protocols would be helpful (including the use of any pill-start that might be permitted).

Response: This has not proven to be a problem to the participating centres to date in the successful recruitment of almost 700 patients and therefore will not be part of this paper. It is however, discussed at length as part of study specific training given to the fertility units by the project management team and is documented in the ES procedure SOP.

Reviewer 2 - Wellington Martins

Q1. Well written protocol assessing an important question.

The only limitation is that the study will be powered only to detect differences > 10% in live birth rate, while a 5% difference (NNT = 20) would already be somewhat relevant, considering the cost/risks of the procedure and the total costs of IVF. However, I do understand that the required sample size to identify a 5% increase would be very large (2900 women should be randomized).

Response: We acknowledge that the study is powered to detect a minimum difference of 10% in live birth rate if it exists. The sample size section provides a rationale for the choice of the 10% difference. As highlighted by the reviewer, to detect a small difference of 5% would require a sample size of 1417 and 1882 per arm (with continuity correction) to preserve a power of 80% and 90% respectively, making the study impractical. With 496 per arm, the study will have a relatively small power of 36.4% to detect a 5%. It should also be noted that the study has been designed with a health economic evaluation. Therefore, if the confidence interval around the observed difference in live birth rates cannot exclude the 5% difference, sensitivity analysis of the cost implications under this scenario will be undertaken and results discussed.

Reviewer 3- Amerigo Vitagliano

General comments made by reviewer: Authors aim to evaluate the effectiveness of mid-luteal endometrial scratching in women undergoing their first IVF-ICSI cycle through a multi-center randomized controlled trial. The topic is of major interest and the study protocol is well designed. As Author correctly state, their study will be the largest randomized controlled trial on patients undergoing their first IVF attempt. Thus, it is expected to provide a powerful contribute for both scientists and fertility care providers about the practice of endometrial scratching.

Given the huge expectations linked to the trial in object, some minor changes to the protocol might be helpful to avoid methodological bias and to implement the reliability of Authors' results.

Response: We thank your reviewer for the comments and would like however to state that the current protocol is version number 5 which has evolved since the beginning of the study and has been through ethical approval. We will give due consideration to the points made and consider whether any further amendments should be made in the future, however we would like in the sake of transparency to publish the protocol as it is at present as it is currently being used to recruit patients.

Q1. Background, page 4, lines 13-34: Authors properly described the results of recent meta-analysis on endometrial injury before IVF, focusing on the findings provided by Nastri et al. In addition, they correctly mention the study by Karimzade et al, emphasizing the negative impact of endometrial scratching at the day of oocyte retrieval on clinical pregnancy rate. For completeness, Authors should specify that current evidence on the effects of endometrial scratching is considerably limited by

heterogeneity among studies in terms of techniques (i.e. curette, pipelle, hysteroscope) and timing (i.e. Nastri et al study includes patients receiving endometrial scratching from the 7th day of the cycle before IVF to the day of oocyte retrieval). Moreover, they should mention that the practice of endometrial scratching is currently being evaluated also in other ARTs (i.e. intrauterine insemination and ovulation induction with sexual intercourse).

Response: Critical analysis of the previous literature, particularly the heterogeneity and the use of endometrial scratch and other non IVF population is of course a valid point for any discussion. In light of this and other reviewer's comments, we have added a sentence regarding such heterogeneity to the background of the manuscript. We feel the mention of scratching in other ARTs is out of the scope of this protocol paper.

Q2. Inclusion criteria, page 8, lines 11-14: "women expected to receive treatment using fresh embryos and considered to be good responders to treatment (Regular ovulatory menstrual cycle, Normal uterine cavity, expected good ovarian reserve)". Authors should state how "normal uterine cavity" was assessed (i.e. with hysteroscopy or sonohysterography) and should provide more details about how ovulation was confirmed (otherwise they may simply state "history of regular menstrual cycle"), as well as criteria for defining "expected good ovarian reserve" (i.e. absence of Bologna criteria, precise cut-off of FSH and/or AMH). Moreover, Authors should specify if autoimmune disease, previous chemotherapy and endocrine disorders will represent additional exclusion criteria.

Response: For the purposes of the manuscript the words were condensed but appreciate the reviewer's comments and will amend this so that the manuscript now includes the full inclusion/exclusion criteria.

Q3. Randomisation, page 9, lines 32-36: "computer is certainly acceptable for generating a random sequence. Nevertheless, a strategy for allocation concealment should be specified in order to avoid bias due to inappropriate allocation (i.e. sealed envelopes, central allocation).

Response: We acknowledge the point raised by the reviewer and apologise for overlooking this point. We have provided the necessary details in the manuscript.

Q4. Trial intervention, page 9, lines 45-46: "A pipelle or similar endometrial sampler is then inserted into the cavity of the uterus". Authors should specify which kind of "endometrial sampler similar to pipelle" will be considered as appropriate in their study, in order to avoid performance bias between centers. Moreover, it would be important to ascertain that "mid-luteal scratching" is performed with comparable timing in all patients. Thus, Authors may clearly define when the procedure will be performed (i.e. from day LH + 6 to LH + 8 if ovulation is assessed by LH measurement, menstrual cycle length minus 5-9 days if ovulation will not be assessed etc.).

Response: I am sure the author would agree the pipelle sampler is the most commonly used device for this procedure. A certain number of samplers maybe reviewed as restrictive and furthermore may promote a potential conflict of interest. Since the study is a pragmatic one and to be able to give an answer which is reproducible across the globe, our intention is to be pragmatic and not introduce any significant changes to the current clinical practice. It is therefore not possible to list all the samplers that can be used commercially, however since we do agree that it is important to know for future validity of the results we are collecting this information when the ES is being performed in our trial database.

Q5. Outcomes, page 11: Authors should provide a clear definition of secondary outcomes (i.e. clinical pregnancy rate will be defined as the visualization of fetal heart activity at ultrasound scan) and the denominator of the outcomes miscarriage rate, ectopic pregnancy rate, multiple pregnancy rate, preterm delivery rate, still birth rate (per patient/per pregnancy).

Response: We appreciate the reviewers' comments and will amend the manuscript to show the full definition of the secondary outcomes.

Q6. IVF cycles: Authors might describe more appropriately the criteria for ovulation induction (i.e. two or more follicles with mean diameter ≥ 18 mm) and specific drugs employed (i.e. 10,000 IU urinary hcg). In addition, as the primary outcome is related to pregnancy, a clear description of luteal phase support, if applicable, (i.e. vaginal progesterone, with dose) should be supplied.

Response: Definition of criteria for triggering oocyte maturation – this study is currently running across 16 centres in the UK, furthermore in order to ensure that the results are applicable across the globe we have kept this as a pragmatic study where again we did not aim to impose any particular changes to local standard operating procedures

Reviewer 4 - Ahmed Gibreel

Q1: This is a very well written protocol to elucidate the genuine effect of endometrial trauma on the outcome of IVF for women undergoing IVF/ICSI for the first time. My only concern about this study related to the population included. Women undergoing IVF for the first time have been consistently shown not to get benefit from the procedure in most of similar trials. A subgroup analysis of trials, where the population was women who underwent ≤ 1 previous attempt of embryo transfer, in the last update of the Cochrane review (Nastri et al,2015). In that analysis, 4 trials were identified and included. Upon aggregating the data, the conclusion was "As the estimate was imprecise, it was not possible to ascertain whether this intervention was related to no effect or benefit: RR 1.1, 95% CI 0.91 to 1.33; P value 0.32; four RCTs; 650 women; $I^2 = 0\%$; low-quality evidence. This means no single randomised trial had shown the very high absolute difference expected in the protocol (increase live birth from 30% to 40 % or an increase in the RR to 1.33). I am a bit concerned that authors may opt to choose this assumption to decrease the sample size needed. Justifications that only procedures that increase success rate by 33% would be worth changing the practice may be too difficult to be accepted. I would recommend recalculating sample size based on the assumption that the procedure would increase Live birth rate from 30% up to 36%.

General comments by reviewer: We thank the reviewer for the comments but regarding the effect of endometrial scratch on first time IVF patients – unfortunately we disagree with the comment made that this has been shown to be ineffective in previous studies. There has been no adequately powered study on endometrial scratch in first time IVF patients, the closest study was that by Young et al which looked at an unselected group of which 70% of them were first time IVF patients but there were significant heterogeneity regarding the management of the IVF cycle and again this was discussed at the point of starting the study with the HTA. Our study will be the first known study to answer this question in this specific homogenous group. Regarding the concerns relating to calculations of the sample size, it is not valid as this is an already funded recruiting study; however the study includes the input from a Professional Statistician at the Clinical Trials Units in Sheffield. The sample size may need to be increased and this is currently undergoing consideration but this is due to possible protocol violations which may have an effect on the intention to treat analysis and not because of the estimated effects of the endometrial scratch.

Q1: In the background section in page 5, line 19: the two references inserted [6,7], I could not figure out why putting one trial and one reference for a protocol published in clinicaltrials as reference for a metaanalysis .

Response: References 6 and 7 have been removed. These two references were the upper and lower bounds of the RR quoted (1.08 and 1.85). As the reviewer points out, these are unnecessary and have been removed

Q2. The authors may want to mention the two large ongoing trials to elucidate the effect of scratch (van Hoogenhuijze et al, 2017) and Lansen et al, 2016).

Response: These trials have been alluded to in the background section of the manuscript.

Reviewer 5 - Carlos Simon

Q1. The methodological design of the paper is OK, however the nature of the complexity of this topic is not due to the lack of clinical data, since more than 1,000 papers, 15 RCTs and 5 meta-analyses have been already published. The main issue to be addressed is the fact that intentional damage to the endometrial lining known as endometrial scratching (ES) has been elevated to the category of therapeutic intervention to improve endometrial receptivity, but ES is not an intervention that has been or can be standardized.

Response: The reviewer points out that over one thousand papers in 15 RCTs and 5 Meta-analysis had been published. We would like to clarify that these papers did not address our current clinical question which refers to a very specific group which has not been adequately addressed in the past in these papers.

We thank the reviewer for his comments regarding the wide use of endometrial scratch in clinical practice before adequate evidence is available. This is exactly the reason why this study is being performed.

Q2: The present study, as all previous, lacks a reproducible definition and methodology of this "intervention" such as the type of catheter to be used, where in the endometrial cavity will be performed, how deep and/or extense the "curettage like" should be, (endometrial tissue obtained from the curettage should be analyzed to determine how deep the 'intervention' reached), US guided or not, at what exact day of the luteal phase will be done and why, etc. Even more worrisome is that the biological responses induced by ES that should be the scientific basis of this practice, remain uncertain and unproven. There is no precedent in any medical field that scratching a tissue or organ without knowing its effects has been elevated to the category of therapeutic intervention and is offering as a "therapy".

Response: The reviewer points to the lack of biological evidence regarding the effectiveness of endometrial scratch. This is indeed a very valid point but is not the topic of this study and is best addressed in another study. Our study is one looking at the clinical effectiveness of an intervention which is already being used in the current population without good evidence. The findings of our study will guide future clinical practice regardless of the logical plausibility as regarding whether or not this procedure should continue to be used or should be immediately discontinued.

The reviewer also points to the lack of homogeneity in clinical practice. Again a valid point, however our study is a pragmatic one and the findings need to be applicable and therefore we are addressing the most common points of clinical practice and not introducing new techniques of standardisation that would not be followed by centres around the world, such as using ultrasound or not, again this is a question that can be answered in a separate study. So whether using the intervention in one particular way is better than another, without knowing that one is superior to another we cannot dictate standard operating procedure for the endometrial scratch across the participating centres.

Minor issues: The background should reflect that no consensus exists about what scratching is, how to perform it and its "therapeutic" effect. Critical literature with the validity of this procedure should also be incorporated for a proper balance.

-Simon C, Bellver J. Scratching beneath 'The Scratching Case': systematic reviews and meta-analyses, the back door for evidence-based medicine. Human Reprod 2014; 29:1618–1621.

- Santamaria X, Katzorke N, Simon C. Endometrial 'scratching': what the data show Current Opinion Obstet Gynecol 4: 242-249, 2016.

Response: These references have been added to the background section

Editorial Requirements:

Q1. Please complete and include a SPIRIT check-list, ensuring that all points are included and state the page numbers where each item can be found: the check-list can be downloaded from here: <http://www.spirit-statement.org>

Response: This has now been included

Q2. Please revise the Strengths and Limitations section (after the abstract) to focus on the methodological strengths and limitations of your study rather than summarizing the results.

Response: These have now been amended in the manuscript

I hope the responses and clarification meet with the editors and reviewers approval and look forward to hearing from the journal as to the outcome of this manuscript.

VERSION 2 – REVIEW

REVIEWER	Sarah Lensen University of Auckland. New Zealand
REVIEW RETURNED	20-Jan-2018

GENERAL COMMENTS	In reply to a number of suggestions the authors state that the requested clarification or information is available in the trial protocol or SOP, or otherwise well known by trial staff or the funder. Unless these documents will be supplied as supplementary files, this does not help the reader, and clarification should be provided in the manuscript. The authors mention they are publishing this protocol for the sake of transparency. Therefore I still think it would be helpful to err on the side of details and have a diagram illustrating the timing of the scratch in relation to the IVF cycle, and to define what exactly is considered a 'normal' AMH etc I couldn't see the SPIRIT checklist that was added? One further point is that participant blinding is not necessarily impossible, and other trials of scratching have implemented sham procedures to this effect. Further, the objectivity of pregnancy as an outcome is not related to participant blinding/performance bias - that would be detection bias (blinding of the outcome assessor) which I agree is unnecessary. Of course these are minor edits and the paper is otherwise well-written and acceptable for publication
---

REVIEWER	Ahmed Gibreel Mansoura University, Egypt
REVIEW RETURNED	25-Jan-2018

GENERAL COMMENTS	I think it is now ready for publication
REVIEWER	Amerigo Vitagliano Department of Women and Children's Health, Unit of Gynecology and Obstetrics University of Padua, Padua, Italy
REVIEW RETURNED	04-Feb-2018
GENERAL COMMENTS	Authors have properly addressed all the issues raised by Reviewers. The present manuscript does not need additional modifications before publication.
REVIEWER	Wellington Martins SEMEAR fertilidade, Brazil
REVIEW RETURNED	06-Feb-2018
GENERAL COMMENTS	Well written protocol for an interesting study.

VERSION 2 – AUTHOR RESPONSE

Reviewer: 1

Reviewer Name: Sarah Lensen

Institution and Country: University of Auckland. New Zealand Please state any competing interests: I have conducted a clinical trial recently on the same intervention and similar patient population; I am aware of this trial and that it is ongoing; I have met the Chief Investigator once before.

Q1. In reply to a number of suggestions the authors state that the requested clarification or information is available in the trial protocol or SOP, or otherwise well known by trial staff or the funder. Unless these documents will be supplied as supplementary files, this does not help the reader, and clarification should be provided in the manuscript. The authors mention they are publishing this protocol for the sake of transparency, Therefore I still think it would be helpful to err on the side of details and have a diagram illustrating the timing of the scratch in relation to the IVF cycle

Response: We thank Dr Lensen for her comments and have amended further to clarify this in the manuscript and noted that the flow diagram (figure 1) includes the wording 'mid-luteal phase'.

Q2. Define what exactly is considered a 'normal' AMH

Response: As this is a pragmatic trial a normal AMH is dictated by the laboratory references range for each centre therefore, we are not able to dictate this centrally.

Q3. One further point is that participant blinding is not necessarily impossible, and other trials of scratching have implemented sham procedures to this effect. Further, the objectivity of pregnancy as an outcome is not related to participant blinding/performance bias - that would be detection bias (blinding of the outcome assessor) which I agree is unnecessary.

Response: The manuscript has been amended to state: 'Since this trial evaluates objectively measured outcomes (pregnancy rates) that are unlikely to be affected by a placebo affect participants will not be blinded to treatment allocation; it is therefore not necessary to perform a sham procedure for the control group. The study statistician, TSC and health economist will be blinded to the allocation'